# Oxidative Stress Biomarkers and Quality of Life Are Contributing Factors of Muscle Pain and Lean Body Mass in Patients with Fibromyalgia

**DOI:** 10.3390/biology11060935

**Published:** 2022-06-19

**Authors:** Jousielle Márcia dos Santos, Ana Cristina Rodrigues Lacerda, Vanessa Gonçalves César Ribeiro, Pedro Henrique Scheidt Figueiredo, Sueli Ferreira Fonseca, Vanessa Kelly da Silva Lage, Henrique Silveira Costa, Vanessa Pereira Lima, Borja Sañudo, Mário Bernardo-Filho, Danúbia da Cunha de Sá Caputo, Vanessa Amaral Mendonça, Redha Taiar

**Affiliations:** 1Brazilian Society of Physiology, Multicentric Postgraduate Program in Physiological Sciences (PPGMCF), Diamantina 39100-000, Brazil; jousielle@hotmail.com (J.M.d.S.); lacerdaacr@gmail.com (A.C.R.L.); vafisio.ribeiro@gmail.com (V.G.C.R.); suffonseca@hotmail.com (S.F.F.); vanessakellysl@hotmail.com (V.K.d.S.L.); vaafisio@hotmail.com (V.A.M.); 2Department of Physiotherapy, Federal University of the Jequitinhonha and Mucuri Valleys, Postgraduate Program in Functional Performance and Rehabilitation (PPGReab), Diamantina 39100-000, Brazil; phsfig@yahoo.com.br (P.H.S.F.); henriquesilveira@yahoo.com.br (H.S.C.); vanessa.lima@ufvjm.edu.br (V.P.L.); 3Department Basic Sciences, Federal University of the Jequitinhonha and Mucuri Valleys, Postgraduate Program in Health Sciences (PPGCS), Diamantina 39100-000, Brazil; 4Department of Physical Education and Sports, Universidad de Sevilla, 41001 Seville, Spain; bsancor@us.es; 5Biophysics and Biometrics Department, Institute of Biology’s Mechanical Vibration Laboratory and Integrative Practices (LAVIMPI), Rio de Janeiro 20021-000, Brazil; bernardofilhom@gmail.com (M.B.-F.); dradanubia@gmail.com (D.d.C.d.S.C.); 6MATériaux et Ingénierie Mécanique (MATIM), Université de Reims Champagne Ardenne, 51100 Reims, France

**Keywords:** oxidative stress, TBARS, superoxide dismutase, lean body mass, fibromyalgia, quality of life, muscle pain

## Abstract

**Simple Summary:**

Fibromyalgia (FM) is a disease that primarily affects women and causes pain all over the body, as well as anxiety, depression, fatigue, weight gain, a decreased quality of life, and difficulties doing daily duties. Although the cause of this disease has yet to be identified, research have been completed or are under underway with the goal of uncovering clues that can explain the disease’s symptoms and proper treatment. Our research looked into whether factors that increase inflammation in the body cause disease symptoms to worsen. Pain, lean mass, quality of life, sleep quality, muscle strength, depression, and probable factors that cause these symptoms to aggravate were assessed in the blood. Women with FM with more pain had a lower quality of life, and women with FM with lower lean mass had muscle weakness in addition to a lower quality of life. Our results recommend that initiatives be implemented to reduce inflammation, improve muscle mass and strength gain and increase the quality of life of these women.

**Abstract:**

(1) The evidence points to an increase in oxygen reactive species as one of the possible causes of fibromyalgia (FM). In addition, it is plausible that an imbalance in redox markers can be associated with pain amplification and dynapenia in FM patients. The aim of our study was to investigate possible factors associated with muscle pain and lean body mass in FM patients. (2) Methods: This was a quantitative, exploratory and cross-sectional study of 47 patients with FM (53.45 + 7.32 years). We evaluated self-perceptions of muscle pain, lean body mass, body composition, quality of life, sleep quality, depression index, muscle performance and oxidative stress biomarkers. (3) Results: We observed that lower blood levels of antioxidants and poor quality of life explained 21% of the greater muscle pain. In addition, high blood levels of oxidative stress, worse muscle performance and poor quality of life explained 27% of the lower lean mass in patients with FM. (4) Conclusions: Larger amounts of lipid peroxidation and reductions in antioxidant levels, in addition to lower muscle performance and poor life quality, are possible independent contributors to greater muscle pain and lower lean body mass in FM patients.

## 1. Introduction

Fibromyalgia (FM) is a chronic disorder marked by widespread muscle pain lasting three months or longer, as well as other symptoms such insomnia, depression, weariness, and exhaustion; pain is reported as the main symptom of FM [1,2]. According to epidemiologic research, the prevalence of FM in the US population is approximately 2–5 percent, while the prevalence of FM in the Brazilian population is around 2 percent [3]. It is worth noting that FM disproportionately affects women (61–90%) [4].

Despite the fact that the pathophysiology of FM is unknown, mounting evidence points to mitochondrial dysfunction as one of the likely causes [5,6], with oxidative stress appearing to be one of the triggering mechanisms of mitochondrial dysfunction [7,8,9]. Mitochondrial abnormalities in subjects with FM are identified as a contributing factor to fatigue and pain [10]. Furthermore, there is a relationship between oxidative stress, pain initiation and FM pathogenesis [11,12,13].

The greater intensity of pain often makes it difficult to perform activities of daily living (ADL), promoting social isolation, worsening the perception of stress and generating a negative impact on the quality of life of these patients [14,15]. Recognizing that social isolation can have negative consequences for one’s overall health, including a deterioration of the physical and mental health, a recent study examined the impact of confining patients with FM and observed that this isolation has a direct and negative impact on the process of central sensitization, aggravating the symptoms of the syndrome [16].

Fibromyalgia symptoms (e.g., pain, stiffness and fatigue) are related to skeletal muscle dysfunction, and investigating muscle changes can shed light on possible explanations for mitochondrial damage. In this context, body composition components such as fat mass, lean mass and other body composition parameters become important points to be evaluated in FM [17]. Furthermore, research suggests that muscle mass loss is ordinary in FM patients and can promote changes in pain, sleep, muscle performance and quality of life [18,19,20].

Of note, FM is often followed by a variety of clinical symptoms such as depression, unrefreshing sleep, irritable bowel syndrome, fatigue and decreased muscle strength [21,22]. Because fibromyalgia can reduce the quality of life of its patients, and that in general patients with FM have difficulty carrying out daily activities [23,24,25], it is relevant to include multiple clinical outcomes in addition to possible markers of oxidative stress, looking for clues of possible outcomes associated with hyperalgesia and dynapenia in FM patients. Therefore, our aim was to investigate the possible association between quality of life (which includes characteristics related to the disease’s impact), muscle performance and oxidative stress markers with skeletal muscle pain and lean body mass in FM patients.

## 2. Materials and Methods

### 2.1. Study Population

The patients were recruited from a waiting list at the Universidade Federal dos Vales do Jequitinhonha e Mucuri’s clinical school of physiotherapy and from advertisements in the Basic Health Units and in the community of Diamantina City, Brazil. All subjects were informed about the study procedures by providing their written consent to participate in this study. The inclusion criteria were the diagnosis of FM confirmed by a rheumatologist, along with being non-smokers and non-consumers of alcohol. The existence of any inflammatory disorders, individuals in psychiatric follow-up and patients receiving oral or topical immune-suppressive drugs were all considered exclusion criteria (corticosteroids). See Appendix A.

### 2.2. Type of Study and Ethical Statement

This is a quantitative, exploratory and cross-sectional study. This work was carried out according to the ethical principles for research involving humans (principles of the Declaration of Helsinki) and received approval from the Ethics Committee of the Universidade Federal dos Vales do Jequitinhonha e Mucuri (No. 4.510.517). All subjects were given written consent to participate in the study after being informed of the methods. The data were collected between April 2017 and June 2018.

### 2.3. Proceedings

The evaluations were performed in the same order every time. Patients arrived at the laboratory at 7 a.m. after fasting for at least 8 h and without taking their medications, as they usually do in the morning. Furthermore, all patients were asked to stop taking their medications for at least 12 h before being evaluated in order to reduce the risk of acute side effects. They had been told to deliver the drugs to the lab. Following the blood collection, standardized food was served and the medicine was administered.

Following the blood collection, a standardized snack was served, and the medicine was administered. Personal and socio-demographic information, as well as a thorough medical history, information on drugs used and information on living habits, was initially obtained on the evaluation form. Following this, the participants went through a preparatory stage that included experimental procedures, instrument application, pain point evaluation, physical testing and anthropometric measures. The total body weight divided by height squared (kg/m^2^) was used to calculate the body mass index (BMI) [26]. The assessments were done by the same investigator to guarantee consistency in instruction.

### 2.4. Independent Variables

The following 8 independent variables (quality of life, depression, sleep quality, muscle performance and four oxidative stress biomarkers) were included as possible independent predictors of muscle pain or lean body mass.

### 2.5. Assessment of the Impact of the Disease on Life Quality

We used the translated and validated FIQ (fibromyalgia impact questionnaire) for the Brazilian population to assess the impact of FM on quality of life. The FIQ is a questionnaire designed to evaluate the impact of FM on life quality through ten components separated into four sub-items: (1) physical variables (functional capacity); (2) psychological disorders (well-being); (3) professional symptoms (absence from work); (4) physical symptoms (difficulties at work, pain, fatigue, stiffness, sleep, anxiety and depression). The FIQ questions are answered using the patients’ self-perceptions from the previous seven days as a reference. The final score ranges from 0 to 100, with a higher number indicating a stronger impact of FM on quality of life [27]. According to studies, scores above 60 indicate a reduced level of intelligence. According to studies, scores above 60 indicate a lower quality of life [28,29,30].

### 2.6. Assessment Sleep Quality

The PSQI (Pittsburgh sleep quality index) was used to measure the patients’ sleep quality during the previous month. The tool permits participants to be classified as ‘excellent sleepers’ or ‘poor sleepers’. There are 19 self-administered questions in the survey. The 19 questions are divided into seven categories, each with a weighting scale ranging from 0 to 3. These scores are then combined together to generate an overall score (0 to 21), with a greater score indicating poorer sleep quality. It is worth noting that a score of five or higher indicates poor sleep quality [31].

### 2.7. Assessment of Depressive Symptoms

The Beck depression inventory (BDI), which has been validated for the Brazilian population, is a 21-item self-assessment questionnaire with four response possibilities (0–3) that assesses attitudes and depressed symptoms. The cut-off points are ≤15 (normal or mild depression), 16–20 (dysphoria) and >20 (depression) [32].

### 2.8. Assessment of Muscle Performance

We also used 5-repetition sit-to-stand tests (5-STS and 60-STS). The patients began by sitting in a chair with arms crossed across their chests and backs against the chair. The seat was roughly 43 cm in height. The researcher stood beside the volunteer, offering directions and preventing the volunteer from falling. A digital timer was used to record the time it took to complete the 5 repetitions [33].

### 2.9. Assessment of Oxidative Stress Biomarkers

The median cubital vein was punctured aseptically to obtain blood samples. To remove cells and debris, tubes containing EDTA were centrifuged at 3000× *g* for 10 min at 20 °C and stored as plasma and erythrocyte aliquots at –80 °C. According to previously published methods, oxidative stress biomarkers were assessed by measuring plasma levels of lipid peroxidation products (thiobarbituric acid reactive substances (TBARS)) [34], enzymatic antioxidants (erythrocyte activity levels of the enzymes catalase (CAT) [35] and superoxide dismutase (SOD) [36]) and non-enzymatic antioxidants (total antioxidant capacity of plasma (FRAP) [37]). The TBARS level was reported in nanomoles MDA per milligram of protein, SOD activity was reported in units (U) per milligram of protein and the CAT activity was reported in DE/min per milligram protein, where DE represents the variation in enzyme activity for 1 min. The total antioxidant capacity was reported as micrograms of FeSO4 per milligram of protein.

### 2.10. Dependent Variables

The following two dependent variables (muscle pain and lean body mass), representing the structural and functional aspects of the International Classification of Functioning, Disability and Health, were considered as dependent variables.

### 2.11. Determination of Muscle Pain

A visual analogue scale (VAS) was used to assess self-reported muscle pain. The patients did this by placing a mark on a 10 cm horizontal line between the two ends, with 0 cm representing no pain and 10 cm representing the worst pain imaginable. Because every pain experience is subjective, the VAS is a subjective scale. This means that factors other than the present pain feeling, such as the current mood, previous experiences and expectations, could influence each pain assessment. Despite this, the VAS is a well-established method of pain assessment, with several advantages—it is simple, effective and widely used in both research and clinical practice. Furthermore, the VAS has been shown to be a reliable tool for assessing both experimental and chronic pain. VAS cut-off points for pain-intensity-related interference with functioning have been suggested for mild (<3.4), moderate (3.5–6.4) and severe pain-related interference (>6.5) [38].

### 2.12. Determination of Lean Body Mass

Dual-energy X-ray absorptiometry: The lean body mass was determined using a densitometer (Lunar Radiation Corporation, Madison, WI, USA, model DPX) [39]. The appendicular lean body mass (ALM) was calculated as the sum of the four members’ muscle mass divided by their height squared (Baumgartner index), representing the lean body mass [40].

### 2.13. Statistical Analysis

GPower software version 3.1.9.2 was used to calculate the sample size. A pilot study with ten FM patients was used to assess the sample size. We calculated 47 FM patients using all dependent variables, an effect size of 0.21 (for muscle pain), eight feasible predictors, a likelihood of error of 5% and 80% power. The Statistical Package for the Social Sciences version 22.0 (SPSS Statistics; IBM, Armonk, NY, USA) and MedCalc Statistical Software version 13.1 (MedCalc Software, Ostend, Belgium) were used to conduct statistical analyses. The Kolmogorov–Smirnov test was used to verify the data distribution. Continuous variables were displayed as the mean and standard deviation (normal distribution) or the median and interquartile range (non-normal distribution), depending on the distribution.

Univariate and stepwise multivariate linear regression were used to confirm the determinants of muscle soreness and lean body mass. In each multivariate model adjusted for age, variables related with muscle pain and lean body mass in the univariate analysis (*p* < 0.1) were included. Four assumptions were used in the linear regression analysis: linearity, residual distribution, homoscedasticity and the absence of multicollinearity. Scatter plots were used to assess the linearity of the independent variables and residuals, and a histogram was used to look at the distribution of residuals. The scatter plot confirmed the homoscedasticity, which was defined by the evenly distributed residuals in the regression line. The variance inflation factor (VIF) values below 10.0 were used to define the absence of multicollinearity. Additionally, the autocorrelation of the variables was verified by the Durbin–Watson test and the values between 1.5 and 2.5 showed that there was no autocorrelation in the data. Statistical significance was set at 5%.

## 3. Results

A total of 71 patients were screened to see if they were eligible. In total, 21 failed to match the inclusion criteria and 3 declined to take part. As a result, 47 patients with FM took part in the study. Table 1 shows the demographics, anthropometrics and clinical features of the subjects.

According to the body mass index (BMI) classification, the patients were overweight (28.99 ± 4.65). The FIQ had a score of 69.69 ± 12.40, indicating poor life quality, while the BDI had a score of (1.06 + 9.64, indicating mild depression. Furthermore, the PSQI score (12.12 ± 3.80) indicated that sleep quality was poor. The average time to complete five repeats during the 5STS (14.68 + 4.04 s) indicated that lower limb function was impaired (Table 1).

In our first analysis model having muscle pain as a dependent variable, we observed that the blood SOD levels were negatively associated while the FIQ score was positively associated with VAS in both univariate and multivariate analyses. In clinical terms, both variables together accounted for roughly 21% of the variation in muscle pain in FM patients (Table 2). The post hoc analysis also revealed a moderate to large effect size (effect size = 0.22; power = 0.80) [41].

In our second analysis model, we found that the TBARS level, FIQ3 score and 5STS were all adversely linked with lean body mass in both univariate and multivariate analyses, accounting for 27% of the variability in lean body mass in FM patients (Table 3). A moderate to high effect size was also discovered during the post hoc analysis (Effect size = 0.24 and Power = 0.80) [41].

## 4. Discussion

In summary, our findings showed that poorer muscle performance, poor quality of life, as well as an imbalance in redox biomarkers may be independent contributing factors to increased muscle soreness and lower lean body mass in patients with FM. In this view, therapeutic efforts to balance redox biomarkers may be one of the priorities of disease management, with an emphasis on improving muscle symptoms in patients with FM, such as muscle pain and lean body mass.

Notably, despite the fact that increased oxidative stress, persistent pain and fatigue are all linked to central sensitization, the mechanism behind its initiation is still unknown [42]. Myalgia and chronic pain are common symptoms of FM, triggering fatigue, inability to perform activities of daily living and as a consequence a negative impact on the quality of life of these patients. Evidence suggests that central sensitization is the underlying mechanism in FM pain; once stress causes muscle pain and an inflammatory response, stress perception can lead to a lower quality of life [10].

Given that mitochondrial dysfunction can cause peripheral muscle pain, weakness, fatigue and exercise intolerance [6], and given that patients with chronic pain frequently have comorbidities such as depression, anxiety, sleep disorders and impaired cognition, as reported in FM [21,22], strategies to modulate chronic pain are becoming more relevant and effective for this population. In their work, Schwartz et al. found a direct link between persistent pain and its reliance on antioxidant activities, particularly SOD [43], which supports our findings that reduced SOD levels are linked to increased muscle pain. Thus, a balance in oxidative stress parameters in people with FM would likely affect blood cytokine levels, reducing muscular soreness and tender points. In this approach, cellular homeostasis disturbances, as well as musculoskeletal oxidative stress and low quality of life, may be predictors of higher muscle pain in FM patients. In addition, we emphasize that the literature indicates that obesity is a major concern for coexistence with FM [14,15]. In our study, the prevalence of overweight and obesity in women with FM was 76% and the mean BMI was 28.99 kg/m^2^. Similar results were previously reported [14,15,16,17,18,19,20]. In addition, a recent study has shown that obese and overweight patients with FM are more impaired than patients with normal weight in terms of pain and quality of life [44], and FM patients with a higher BMI have a lower lean mass and worse quality of life [19].

The European Working Group on Sarcopenia in Older People (EWGSOP) has published a clinical definition and diagnostic criteria for sarcopenia, as well as population-based threshold values [45]. The (EWGSOP) recommends a lean body mass cut-off point of less than 5.5 kg/m^2^ for sarcopenia [46]. In this regard, our sample was classified as having dynapenia based on age and sex, despite the fact that FM patients were non-sarcopenic (lean body mass about 7.17 kg/m^2^). This conclusion is in line with earlier research revealing that patients with FM have a lower lean body mass than age-matched healthy people [47,48]. Aside from the fact that our sample had dynapenia, the lower limb function (approximately 14.68 s) was lower than the clinically significant cut-off for sex and gender, i.e., 11.4 s [49,50]. In this context, earlier research by our group found that patients with FM had lower extremity functionality that was worse than healthy matched (sex and years) controls [11,29].

In addition, there is evidence that altered oxidative stress indicators are linked to the severity of FM, including physical symptoms (muscle soreness and fatigue), as well as to poorer muscle performance and quality of life [11,16]. Thus, a musculoskeletal cell homeostasis imbalance can have an impact on quality of life, including on mental health [51].

The FIQ measures the impacts of FM, including physical factors, psychological disorders, the professional situation and physical symptoms. As a result, our study revealed that sub-item 3, i.e., professional situation as measured by the number of days missed from work due to the disease’s influence on these patients’ lives, had a stronger impact on daily activities, which worsens the quality of life of FM patients [30]. In fact, our data showed that a greater number of absences from work is a determinant of having lower lean body mass in this population.

The potential impacts of oxidative imbalance, poor functioning and diminished quality of life on muscle pain and lean body mass suggest that all of these parameters should be investigated in the context of FM. As a result, more research into the cause–effect correlations and prognostic factors of FM, such as mitochondrial dysfunctions, pain, disease impacts and oxidative stress in FM patients, is needed. Second, because this study focused just on women, it is important not to extrapolate too far from the findings. Finally, because blood levels of oxidative biomarkers appear to interact with age, our findings cannot be applied to other populations (e.g., other gender and age groups).

The present investigation had the advantage of being conducted under controlled and uniform conditions. Furthermore, our research is groundbreaking because no previous research has looked into the possible link between oxidative stress parameters and the disease’s influence on widespread muscle pain and lean body mass in FM patients.

## 5. Conclusions

The findings of this study were clinically relevant, demonstrating that lower muscle performance and poor life quality, as well as a reduction in superoxide dismutase levels and an imbalance favoring the degradation of lipid peroxidation byproducts, are possible independent contributors to greater widespread muscle pain and lower lean body mass in FM patients.

## Figures and Tables

**Table 1 biology-11-00935-t001:** Characteristics of fibromyalgia patients (*n* = 47).

Variables	
Age (years)	53.45 ± 7.32
BMI (kg/m^2^)	28.99 ± 4.65
Lean body mass (kg/m^2^)	7.17 ± 0.90
Muscle pain (VAS)	6.17 ± 2.18
FIQ (score)	69.69 ± 12.40
5STS (seconds)	14.68 ± 4.04
BDI (score)	21.06 ± 9.64
PSQI (score)	12.12 ± 3.80
TBARS (nmol MDA/mg protein)	0.47 ± 0.22
CAT (U/mg protein)	2.99 ± 1.44
SOD (U/mg protein)	32.51 ± 9.42
FRAP (FeSO4.1^−1^ mg protein^−1^)	278.66 ± 108.79

Data presented as mean ± SD. Abbreviations: BMI: body mass index; 5STS: sit to-stand test; BDI: Beck depression inventory; PSQI: Pittsburgh sleep quality index FIQ: fibromyalgia impact questionnaire; TBARS: substances reactive to thiobarbituric acid; CAT: catalase; SOD: superoxide dismutase; FRAP: ferric reducing antioxidant power.

**Table 2 biology-11-00935-t002:** Independent contributors to muscle pain in fibromyalgia patients (*n* = 47).

Independent Variables	± SE	Univariate		Multivariate				*p* Value
		R^2^ Adjusted	Beta	*p* Value	± SE	R^2^ Adjusted	Beta	
FIQ (score 0–100)	0.06 ± 0.02	0.12	0.35	0.01 *	0.05 ± 0.02	0.21	0.30	0.02 *
SOD (U/mg protein)	−0.09 ± 0.03	0.16	−0.40	0.05 *	−0.0 ± 0.03		−0.36	0.009 *
BDI (score 0–63)	0.08 ± 0.03	0.12	0.34	0.01 *	0.05 ± 0.02			NS
TBARS (nmol MDA/mg protein)	−2.11 ± 1.44	0.04	−0.21	0.15	0.05 ± 0.02			NS
5STS (seconds)	0.13 ± 0.07	0.05	0.24	0.10	0.05 ± 0.02			NS
BMI (kg/m^2^)	0.11 ± 0.06	0.06	0.25	0.08	0.05 ± 0.02			NS
Lean body mass	0.51 ± 0.35	0.04	0.21	0.15	0.05 ± 0.02			NS
Age (years)	0.009 ± 0.01	0.01	0.07	0.62	0.05 ± 0.02			NS

Independent contributors to muscle pain in fibromyalgia patients (*n* = 47): beta coefficient; SE: standard error; R^2^ adjusted: adjusted coefficient of determination. Abbreviations: FIQ: fibromyalgia impact questionnaire; VAS: analogic visual scale; BDI: Beck depression inventory; 5STS: sit-to-stand test; TBARS: substances reactive to thiobarbituric acid; SOD: superoxide dismutase; NS: non-significance. Note: *p*-value < 0.05.

**Table 3 biology-11-00935-t003:** Independent contributors to lean body mass in fibromyalgia patients (*n* = 47).

Independent Variables	± SE	Univariate		Multivariate				*p* Value
		R^2^ Adjusted	Beta	*p* Value	± SE	R^2^ Adjusted	Beta	
FIQ3 (score 0–100)	−0.10 ± 0.03	0.12	−0.39	0.04 *	−0.08 ± 0.03	0.27	−0.32	0.01 *
5STS (seconds)	−0.06 ± 0.03	0.16	−0.27	0.05 *	−0.06 ± 0.03		−0.28	0.04 *
TBARS (nmol MDA/mg protein)	−1.32 ± 0.58	0.12	−0.32	0.02 *	−1.53 ± 0.53		−0.37	0.006 *
FRAP (FeSO4.1^−1^ mg protein^−1^)	−0.02 ± 0.01	0.04	−0.28	0.01 *				NS
CAT	−0.12 ± 0.09	0.05	−0.19	0.19				NS
SOD	−0.02 ± 0.01	0.06	−0.21	0.15				NS
Age (years)	0.009 ± 0.01	0.04	0.07	0.62				NS

Independent contributors to lean body mass in fibromyalgia patients (*n* = 47): beta coefficient; SE: standard error; R^2^ adjusted: adjusted coefficient of determination. Abbreviations: 5STS: sit-to-stand test; FIQ3: fibromyalgia impact questionnaire—professional situation; TBARS: substances reactive to thiobarbituric acid; NS: non-significance. Note: *p*-value < 0.05.

## Data Availability

The study data are with the researchers and can be provided when necessary.

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
