# Peer review of "Oxidative Stress Biomarkers and Quality of Life Are Contributing Factors of Muscle Pain and Lean Body Mass in Patients with Fibromyalgia"

_biology, 2022, doi:10.3390/biology11060935_

Round 1

Reviewer 1 Report

Thank you for an interesting manuscript. 

I have mainly assess the statistical methods and other reviewers need to assess the medical significance of the work.

Some minor comments on the statistical analysis:

* Please designate the models as univariable and multivariable instead of univariate and multivariate.

*Please provide more information about the stepwise procedure, e.g. cut-off criteria for including or excluding variables.

* Please add also the unstandardized coefficients with confidence intervals in the tables in addition to the standardized coefficients (Beta)

Author Response

REVIEWER 1

Comments and Suggestions for Authors:

In presente studie authors aims to show and verify their hypothezis about the relation of Fibromyalgia progresis and poor quality of life and an imbalance in biomarkers of oxidative stress. Although the impact of the disease and an imbalance in biomarkers of oxidative stress is well known, prestented point of view is interesting but I am afraid that the presented dependencies are unfortunately too simplified.

We thank the Reviewer for appreciating the potential impact of our work and for the comments and suggestions that provided a significant improvement of the new version of the manuscript.

The authors did not attempt to present the mechanisms of the development of presented disorders and the severity of inflammation process and why it may be associated. Furthermore it should be metioned that in the case of such diseases, its severity and the symptoms are the underlying causes of the poor quality of life.

There are some limitations in the studie: authors should add souch paragraph in the text to sum up whole analyzes and presente it clearly.

(Page 2 - Lines: 49).

Authors should add some interpretation abouthe the BMI (presented values are high ...28.99 ± 4.65) and show the relationship with lean body mass. Than we can conclude other aspects.

(Page 11 - Lines: 283).

Reviewer 2 Report

In presente studie authors aims to show and verify their hypothezis about the relation of Fibromyalgia progresis and poor quality of life and an imbalance in biomarkers of oxidative stress. Although the impact of the disease and an imbalance in biomarkers of oxidative stress is well known, prestented point of view is interesting but I am afraid that the presented dependencies are unfortunately too simplified. 

The authors did not attempt to present the mechanisms of the development of presented disorders and the severity of inflammation process and why it may be associated. Furthermore it should be metioned that in the case of such diseases, its severity and the symptoms are the underlying causes of the poor quality of life. 

There are some limitations in the studie: authors should add souch paragraph in the text to sum up whole analyzes and presente it clearly. 

Authors should add some interpretation abouthe the BMI (presented values are high ...28.99 ± 4.65) and show the relationship with lean body mass. Than we can conclude other aspects. 

Author Response

REVIEWER 2

Comments and Suggestions for Authors:

The authors have developed an interesting and useful work on patients affected by fibromyalgia to verify if there could be a relationship between a poor quality of life that includes features related to the impact of the disease and an imbalance in oxidative stress biomarkers. However, I would like to make some observations before recommending your work for publication.

We thank the Reviewer for appreciating the potential impact of our work and for the comments and suggestions that provided a significant improvement of the new version of the manuscript.

I recommend that the authors detail in the "Abstract" the type of study that was carried out.

Response: We thank the Reviewer for this observation. We have included the type of study in the abstract in the revised version of the manuscript.

In the context of a COVID-19 pandemic, I recommend that the authors comment on the situation by citing the following paper: DOI: 10.29333/ejgm/11798.

Response: We thank the Reviewer for the suggestion. We included a paragraph and inserted the suggested reference.

(Page 2 - Lines: 57).

Would it be possible that the authors could provide some graphical figures? It would help to better understand the manuscript.

Response: We appreciate the Reviewer's request and inserted an graphical abstract (see below) in the article. However, we opted to present the data using Tables because are the most effective way to show data for reference, and they may hold a lot of information. Using an appropriate label and showing the data in relevant categories, the provided data may be effectively interpreted (sorted in columns and rows).

Reviewer 3 Report

The authors have developed an interesting and useful work on patients affected by fibromyalgia to verify if there could be a relationship between a poor quality of life that includes features related to the impact of the disease and an imbalance in oxidative stress biomarkers

However, I would like to make some observations before recommending your work for publication.

  1. I recommend that the authors detail in the "Abstract" the type of study that was carried out.
  2. In the context of a COVID-19 pandemic, I recommend that the authors comment on the situation by citing the following paper: DOI: 10.29333/ejgm/11798
  3. Would it be possible that the authors could provide some graphical figures? It would help to better understand the manuscript.

Author Response

REVIEWER 3

Comments and Suggestions for Authors

The authors have developed an interesting and useful work on patients affected by fibromyalgia to verify if there could be a relationship between a poor quality of life that includes features related to the impact of the disease and an imbalance in oxidative stress biomarkers. I would like to make some observations before recommending your work for publication.

We thank the Reviewer for appreciating the potential impact of our work and for the comments and suggestions that provided a significant improvement of the new version of the manuscript.

I recommend that the authors detail in the "Abstract" the type of study that was carried out.

Response: We thank the Reviewer for this observation. We have included the type of study in the abstract in the revised version of the manuscript.

In the context of a COVID-19 pandemic, I recommend that the authors comment on the situation by citing the following paper: DOI: 10.29333/ejgm/11798.

Response: We thank the Reviewer for the suggestion. We included a paragraph and inserted the suggested reference.

(Page 2 - Lines: 57).

Would it be possible that the authors could provide some graphical figures? It would help to better understand the manuscript.

Response: We appreciate the Reviewer's request and inserted an graphical abstract in the article (see below). However, we opted to present the data using Tables because are the most effective way to show data for reference, and they may hold a lot of information. Using an appropriate label and showing the data in relevant categories, the provided data may be effectively interpreted (sorted in columns and rows).

Round 2

Reviewer 3 Report

Congratulations to the authors for their work.

Kind regards,